# Hallucination Detection and Mitigation with Diffusion in Multi-Variate Time-Series Foundation Models

**Vijja Wichitwechkarn**
*Department of Engineering*
*University of Cambridge*

*vw273@cam.ac.uk*

**Charles Fox**
*School of Computer Science*
*University of Lincoln*

*ChFox@lincoln.ac.uk*

**Ruchi Choudhary**
*Department of Engineering*
*University of Cambridge*

*rc488@cam.ac.uk*

**Reviewed on OpenReview:** *https://openreview.net/forum?id=fHGQ7hZlb5*

## Abstract

Foundation models (FMs) for natural language processing have many coherent definitions of hallucination and methods for its detection and mitigation. However, analogous definitions and methods do not exist for multi-variate time-series (MVTS) FMs. We propose new definitions for MVTS hallucination, along with new detection and mitigation methods using a diffusion model to estimate hallucination levels. We derive relational datasets from popular time-series datasets to benchmark these relational hallucination levels. Using these definitions and models, we find that open-source pre-trained MVTS imputation FMs relationally hallucinate on these datasets on average up to 59.5% as much as a weak baseline. The proposed mitigation method reduces this by up to 47.7% for these models. The definition and methods may improve adoption and safe usage of MVTS FMs.

## 1 Introduction

Foundation models (FMs) trained on large and diverse datasets, that can be prompted to perform many types of computation, have enjoyed rapid progress in Natural Language Processing (NLP). Examples include Llama Touvron et al. (2023), ChatGPT Achiam et al. (2023), Claude Min et al. (2023) and Gemini Anil et al. (2023). Models with similar capabilities are now also seen in other domains including time-series modelling. Recent works have shown that pre-trained models for time-series forecasting can be used effectively on unseen forecasting domains in a zero-shot manner. This is achieved by training on large quantities of time-series data from diverse domains as in Chronos Ansari et al. (2024), TimesFM Das et al. (2023), LagLlama Rasul et al. (2023), TimeGPT Garza & Mergenthaler-Canseco (2023), MOIRAI Woo et al. (2024). Similar models have also been successful for time-series imputation such as MOMENT Goswami et al. (2024), TIMER Liu et al. (2024), TOTEM Talukder et al. (2024), TimesNet Wu et al. (2022) and GPT4TS Zhou et al. (2023).

We argue that pre-trained models for multi-variate time-series (MVTS) imputation are closer than MVTS forecasting to what are typically referred to as FMs in NLP as these can be prompted to handle different tasks. Prompts are the provided values and responses are the imputed values. For example, forecasting can be prompted for by masking future time-steps and asking the model to fill in these masked values; while interpolation can be prompted for with data from both before and after the missing period. Imputation therefore provides an interface for arbitrary question answering in MVTS. This work will therefore focus on

these models, particularly MOMENT Goswami et al. (2024) and TIMER Liu et al. (2024) which are the only FMs of this type that currently have open-source weights available.

For MVTS question answering to be useful in real-world cases, a measure of confidence in the model's response is required, analogous to hallucination detection in NLP. To our knowledge, there is no literature on hallucination definition and detection in MVTS imputation, even with the advent of MVTS FMs. This is in stark contrast to NLP, where a large and active body of work exists on defining, categorizing, and detecting different types of hallucinations Rawte et al. (2023); Zhang et al. (2023c); Ye et al. (2023). Much like in NLP, where new definitions have emerged and pre-existing concepts were unified under the umbrella of 'hallucination', we argue that the MVTS research would benefit similarly from this unification. This is especially true with the increasing transfer of concepts from NLP such as FMs.

The contributions of this work include the definition of two types of hallucination in the context of MVTS imputation: distributional and relational. These are defined using established definitions from the NLP literature. Distributional hallucination is grounded by a pre-existing concept in the MVTS literature while relational hallucination is a new concept, which is the main focus of this work. We use diffusion models Ho et al. (2020) for MVTS imputation and propose a method to detect and mitigate hallucination in its response. We also show that MVTS FMs hallucinate heavily using popular MVTS datasets, and that this can be detected and mitigated using our proposed methods. Project repository: https://github.com/vijja-w/mvts-rhallu.

## 1.1 Diffusion Model Preliminaries

Diffusion models are probabilistic generative models that iteratively degrade data by introducing noise, then learn to reverse this process. This allows them to iteratively generate new samples by sampling from a simple prior, which is typically a Gaussian distribution Yang et al. (2023). They have become well known in image generation Rombach et al. (2022) and have been applied extensively to various fields including time-series generation Yuan & Qiao (2024), forecasting Meijer & Chen (2024) and imputation Wang et al. (2024); Yang et al. (2024b). Ever since diffusion models have been applied to time-series imputation Tashiro et al. (2021), there has been growing work to improve them for this use case. These include improvements to the masking criteria during training Xiao et al. (2023); Chen et al. (2023b); Liu et al. (2023), the architectures used Alcaraz & Strodthoff (2022) and the sampling process Wang et al. (2023). Diffusion models have since become widely popular, becoming one of the best performing methods for time-series imputation Zhou et al. (2024). The background on diffusion models that is directly used in this work will be explained in the following sections. This includes the mathematical notations for Denoising Diffusion Probabilistic Models (DDPM) Ho et al. (2020) and conditioning through RePaint Lugmayr et al. (2022).

### 1.1.1 Unconditional Diffusion Models

In the forward process, samples from the training data $x_0$ are increasingly corrupted through the addition of Gaussian noise for $T$ time-steps to generate noisy samples $x_1, ..., x_T$:

$$q(x_t|x_{t-1}) := \mathcal{N}\left(x_t; \sqrt{1-\beta_t}x_{t-1}, \beta_t \mathbf{I}\right) \tag{1}$$

$$q(x_{1:T}|x_0) := \prod_{t=1}^{T} q(x_t|x_{t-1}). \tag{2}$$

The Gaussian noise is determined by the variance schedule $\beta_1, ..., \beta_T$ which is typically linearly increasing. The forward process also admits sampling timestep $t$ directly:

$$q(x_t|x_0) = \mathcal{N}\left(x_t; \sqrt{\bar{\alpha}_t}x_0, (1-\bar{\alpha}_t)\mathbf{I}\right), \tag{3}$$

where $\alpha_t := 1 - \beta_t$ and $\bar{\alpha}_t := \prod_{s=1}^{t} \alpha_s$.

The reverse process is used to successively denoise the corrupted data by learning $p_\theta(x_{t-1}|x_t)$ using a neural network with learnable parameters $\theta$:

$$\mu_\theta(x_t, t) = \frac{1}{\sqrt{\alpha_t}}\left(x_t - \frac{1 - \alpha_t}{\sqrt{1 - \bar{\alpha}_t}}\epsilon_\theta(x_t, t)\right), \tag{4}$$

$$p_\theta(x_{t-1}|x_t) := \mathcal{N}\left(x_{t-1}; \mu_\theta(x_t, t), \Sigma(x_t, t)\right), \tag{5}$$

$$p_\theta(x_{0:T}) := p(x_T)\prod_{t=1}^{T} p_\theta(x_{t-1}|x_t), \tag{6}$$

where $\mu_\theta(x_t, t)$ is the predicted mean used to sample $x_{t-1}$ and $\epsilon_\theta(x_t, t)$ is the noise predicted by a neural network at time-step $t$. The original DDPM Ho et al. (2020) sets $\Sigma_\theta(x_t, t) = \sigma_t^2\mathbf{I}$, where $\sigma_t^2 = \beta_t$. We will use this DDPM formulation of diffusion models in this work due to its popularity and simplicity.

### 1.1.2 Conditioning Diffusion with RePaint

Diffusion models as described above are referred to as unconditional diffusion models as they do not directly allow for conditioning to be applied. It is however possible to guide the unconditional diffusion model using the RePaint Lugmayr et al. (2022) method. Here, components of the input vector $x$ are split into conditioning values $x^{(c)}$ and missing values $x^{(m)}$ to be imputed by the model. The missing values are sampled in the same way as Eq. 5:

$$p_\theta(x_{t-1}^{(m)}|x_t^{(m)}) := \mathcal{N}\left(x_{t-1}^{(m)}; \mu_\theta(x_t^{(m)}, t), \Sigma(x_t^{(m)}, t)\right). \tag{7}$$

The conditioning values however uses the corrupted version obtained using Eq. 3:

$$q(x_t^{(c)}|x_0^{(c)}) = \mathcal{N}\left(x_t^{(c)}; \sqrt{\bar{\alpha}_t}x_0^{(c)}, (1 - \bar{\alpha}_t)\mathbf{I}\right). \tag{8}$$

In this way, at each time-step $t$ of the reverse process, $x_t$ is composed of the imputed missing values $x_t^{(m)}$ obtained by denoising, and the conditioning values $x_t^{(c)}$ obtained by corrupting the actual given values to the correct noise level associated with the time-step $t$. The predicted mean at each diffusion time-step is then computed as usual using Eq. 4.

We will use RePaint to condition an unconditional diffusion model trained on our dataset and impute missing values. This allows for arbitrary question answering using the prompt-response framework at inference without modification to the training procedure of the diffusion model.

### 1.2 Hallucination Detection and Mitigation in NLP

Example methods for hallucination detection and mitigation in NLP include the use of external knowledge retrieved from the web or task-specific databases to identify and correct non-factual content in responses Peng et al. (2023); Shuster et al. (2021); Lewis et al. (2020); Chen et al. (2023a); Varshney et al. (2023). However, effective knowledge retrieval can be challenging and costly to run in practice Mündler et al. (2023). It is unclear how this transfers to MVTS as there are no clear-cut facts to retrieve. There are also methods that do not use external knowledge but instead uses multiple samples from the same prompt to measure consistency of the generated information Manakul et al. (2023); Elaraby et al. (2023); Zhang et al. (2023a); Farquhar et al. (2024). This can also be done using an ensemble of models Du et al. (2023). Similarly to these methods, this work will mitigate hallucination through sampling. The concept of consistency, however, does not transfer to MVTS as these require clear-cut facts and contradictions. A separate hallucination detection model can also be trained to detect hallucination from the generated text Chen et al. (2023c); Pacchiardi et al. (2023); Mishra et al. (2024); Zha et al. (2023) or the model's internal states Su et al. (2024). This is the approach that will be adopted in this work using a diffusion model. There has also been work on scaling the generation of datasets that can be used to train these models Su et al. (2024); Gu et al. (2024). Hallucination mitigation can also be achieved through direct supervised finetuning Gu et al. (2025); Tian et al. (2023); Lin et al. (2024); Zhang et al. (2024); Chen et al. (2024). However, the fine-tuned model still has to be used in conjunction with hallucination detection methods since they can still hallucinate, albeit at a potentially reduced rate.

## 2  Defining Hallucination for Multi-Variate Time-Series Imputation

There is a large and active literature on defining, detecting and mitigating hallucination in NLP. In this context, hallucination is commonly defined as the behaviour when models generate responses with information that is *false* Rawte et al. (2023); Zhang et al. (2023c); Ye et al. (2023). In time-series however, there are no clear-cut facts as in language. Consequently, there is no absolute truth to time-series, only what is probable relative to the provided context dataset. We therefore define **distributional hallucination** as a type of hallucination in time-series where the combination of the prompt and the generated response is out of distribution (OOD) with respect to a target dataset. Note that if an OOD prompt is provided to a model, all responses will automatically be classified as a distributional hallucination. This is important in the context of FMs trained on large quantities of data since it is typically unknown whether a prompt is OOD or not. In practice, distributional hallucination is a continuous concept, so a threshold must be chosen to define a prompt-response pair as distributionally hallucinating.

Another definition of hallucination in NLP is the generation of *self-contradictory* responses Mündler et al. (2023); with incoherent explanation and reasoning Zhang et al. (2023b); or responses that are irrelevant to the prompt Gallifant et al. (2024). These definitions will be used as the NLP analogue of what will be referred to as **relational hallucination**. A relation between a set of $N$ variables $\mathbf{x} = \{x_1, x_2, ..., x_N\}$ can be written as $f(\mathbf{x}) = 0$, where $f$ is some ground-truth function that defines the relation. The 'relational error' which measures the degree of which the relation is broken can then be defined as $E_r := |f(\mathbf{x})|$. Relational hallucination can then be defined as the case when the model returns a set of variables that has 'high' relational error, relative to some threshold. This occurs when the prompt and the response are incompatible, given $f$. This is analogous to a response that is irrelevant to the prompt in the NLP case. Additionally, relational hallucination can occur when the variables returned in the response are incompatible with themselves. This case is analogous to self-contradiction in NLP hallucination. Incoherent explanations and reasoning can also be seen as a form of self-contradiction. In the same way as distributional hallucination, relational hallucination is also defined relative to a given dataset.

**Examples**  In contrast to distributional hallucination, an OOD prompt may not necessarily result in relational hallucination. As a concrete example, consider the following case with three variables $\{x_0, x_1, x_2\}$ where the ground truth relation is addition: $x_0 + x_1 - x_2 = 0$. The training dataset consists of $x_0, x_1 \in [0, 10]$ and $x_2 \in [0, 20]$. The combination of the prompt, where $x_0 = 21$ and $x_1 = 22$, and the response $x_2 = 43$, will be classified as distributionally hallucinating but not relationally hallucinating. The combination of the prompt, where $x_0 = 1$ and $x_1 = 2$, and the response $x_2 = 7$, will be classified as both distributionally hallucinating and relationally hallucinating. In this sense, relational hallucination is a subset of distributional hallucination.

**Relational Hallucination is More Important**  Distributional hallucination is important for detecting whether a question is OOD, which is typically not known at inference. Relational hallucination also captures all the in-distribution data, since by definition they all have correct relations between the variables. They however also extend to regions of the state space that is OOD. In this sense, relational hallucination is less restricted than distributional hallucination. Models being able to generalise to and operate in regions which are OOD is important as a large family of important question types are OOD. For example, to optimise variables to achieve better performances given the current data or to simulate a system under new conditions. This work will therefore focus on relational hallucination.

**Related Concepts**  *OOD detection* aims to detect test samples that do not exist in the training distribution Yang et al. (2024a). This is what we refer to as distributional hallucination in our work. We adopt the term 'hallucination' as this is the common term with respect to FMs, and also because it has proven useful to place pre-existing concepts under the same umbrella to consolidate definitions as seen in NLP. *Anomaly Detection* in contrast aims to detect unusual cases which may exist in the training set Zamanzadeh Darban et al. (2024), assuming that the majority of training data is from the 'correct' distribution and a minority of data is from an 'anomalous' distribution. Anomaly detection can therefore be seen as OOD detection but with the definition of being 'in distribution' replaced with being in the 'correct distribution'. Anomaly detection in MVTS predicts which time indices within a single MVTS window correspond to anomalous

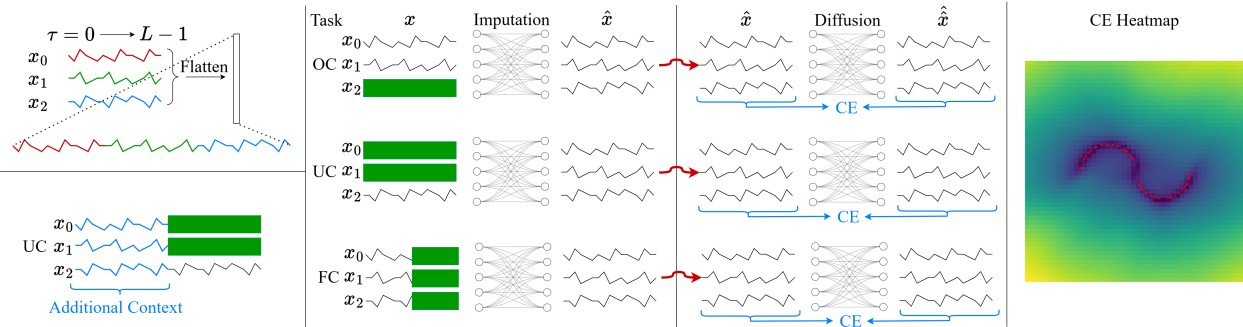

Figure 1: (**Left Top**) Flattening a datapoint with three variables. **(Left Bottom)** Example of additional context provided on the UC task. **(Middle Left)** Different type of tasks (OC, UC and FC) for the prompt. Masked variables are shown as blank green boxes and unmasked variables are used as the prompt. The imputation process can be done using the diffusion model or pre-trained foundation models. **(Middle Right)** The combination of the prompt and response obtained from the imputation is used as the prompt for a diffusion process, which is used to compute Combined Error (CE) metric. (**Right**) CE heatmap for a diffusion model fit to a small non-linear 2D dataset. Red dots are the data points in the training set and darker colours correspond to lower CE values.

values. Relational hallucination differs from these definitions, as it measures the compatibility of all the values in a MVTS window. A MVTS window can be out of distribution but still be relationally correct. Relational hallucination is a new concept in MVTS that is transferred from NLP for MVTS FMs. It is the main focus of this work.

## 3 Relational Hallucination Detection and Mitigation using Diffusion Models

Previous works have shown that diffusion models trained to generate images can detect hallucinations in their generated outputs Aithal et al. (2024). They have also been successfully applied to MVTS imputation Zhou et al. (2024) and anomaly detection Chen et al. (2023b). We therefore consider diffusion models a promising candidate for arbitrary MVTS question answering through imputation, and the detection of relational hallucination.

**Notations**  To describe the prompt-response framework for MVTS imputation, we will use the following notation for each data point: $x_i$, where $i \in \mathcal{I}$ indexes the data dimension. A prompt is defined by specifying the set of variables that will be used as the prompt $i \in \mathcal{I}_\mathrm{p}$ and setting their values accordingly. The values for the remaining indices $\mathcal{I}_\mathrm{r} = \mathcal{I} \setminus \mathcal{I}_\mathrm{p}$ will be masked and imputed by the model to generate the response. As before, the predicted mean at each diffusion time-step will be denoted $\mu_{i,t}$ where $t \in \{0, ..., T\}$. Note that denoising decrements the time-step from $T$ to $0$. The final output (prediction) from the model ($t = 0$) will be denoted as $\hat{x}_i$, where the imputed response is $\hat{x}_i, \forall i \in \mathcal{I}_\mathrm{r}$, and $\hat{x}_i \approx x_i, \forall i \in \mathcal{I}_\mathrm{p}$.

**Conditioning**  Once the prompt is defined, RePaint Lugmayr et al. (2022) is used to condition an unconditional diffusion model trained on the dataset. The prompt is used as the conditioning $x^{(c)}$ in RePaint as described in Section 1.1.2. This allows diffusion models to act as a prompt-response model for general time-series question answering.

**Relational Hallucination Metric**  We propose the Combined Error (CE) metric that can be used to estimate the level of relational hallucination and a method to extract it from a diffusion model trained on the dataset. This metric can be computed for a given prompt-response pair $\hat{x}_i, \forall i \in \mathcal{I}$ obtained from some model such as a FM. It is computed by using RePaint to condition the diffusion model and setting the prompt as $x_i := \hat{x}_i, \forall i \in \mathcal{I}$. The output of this process will be referred to as $\hat{\hat{x}}_i$ where the double hat denotes

a prediction where the target is a previous prediction. The CE metric can be computed as

$$M_{\text{CE}} = \text{RMSE}_i(\hat{\hat{x}}_i, \hat{x}_i), \ \forall i \in \mathcal{I}, \tag{9}$$

where the root mean square error is taken across the data dimension. Note that $\hat{\hat{x}}_i$ can be computed using a single denoising step (the final time-step going $t = 0$). This is because RePaint allows the diffusion process to be skipped to the final step for all conditioning values, which in this case is the entire data dimension. This is done using the forward process (Eq. 8). Denoising using Eq. 4 is therefore only done on the final diffusion time-step and obtaining this metric is not computationally expensive. This process is shown in Fig. 1 (middle right).

To highlight properties of the CE metric, a diffusion model was trained on a small nonlinear 2D dataset. Fig. 1 (right) visualises the value of the CE metric for each point in this space. The CE metric is low in regions where the relations hold. This is true even in OOD regions without data. We also tested variations of metrics similar to Aithal et al. (2024). However, these are not effective, as shown in Appendix A.

**Hallucination Detection**  To use the proposed CE metric to gauge the expected level of relational hallucination, a dataset-specific scale is required to determine whether the metric is high/low relative to the dataset. We propose a simple method. Firstly, the CE metric is obtained for all the prompt-response pairs obtained from the training set, over all the imputation tasks. The quartiles of the CE metric are then computed. These quartiles are then used to classify the prompt-response pairs into classes of expected relational hallucination levels at inference: low (below the second quartile), medium (between the second and third quartile) and high (greater than the third quartile).

**Hallucination Mitigation**  we also propose a simple method for mitigating relational hallucination for non-deterministic models. For a given prompt, $N$ responses are sampled from the model. The CE metric can then be computed for all the obtained prompt-response pairs $\left(\hat{x}^{(j)}, M_{CE}^{(j)}\right) \in \mathcal{X}_{\text{sampled}}$. The prompt-response pair with the lowest metric $\hat{x}^{(j^*)}$, where $j^* = \arg\min\left(M_{CE}^{(j)}\right)$, is then selected as the response with the expected lowest relational hallucination.

## 4    Experiments

**Evaluation Method**  In real-world settings, the ground-truth relation function $f$ is typically unknown. So, the ground-truth relational error cannot be computed for a given prompt-response pair. This work proposes the use of diffusion models to generate a metric for a given prompt-response pair, that can be used as an indirect measure for the relational error, and hence relational hallucination. To evaluate our proposed methods however, we require a known $f$ that can be used to compute the ground-truth relational error. To achieve this, we can add 'relational variables' to a dataset, which has a known relation $f$ with other variables, and use this to compute the ground-truth relational error for evaluation. Since our method has to model all the variables together as a joint distribution, it does not have access to this ground-truth relation $f$. This effectively allows us to evaluate our method in the real-world situation where the ground-truth $f$ is not known. We apply this procedure to popular MVTS datasets.

**Relational Variables**  We add relational variables to popular MVTS datasets and refer to these datasets by prefixing 'r' to their names. The relational variable added to the Electricity Consuming Load (rECL) dataset Trindade (2015) is the difference between two other variables. The relational variable added to the Weather dataset (rWTH) Max Planck Institute for Biogeochemistry (2024), is added by computing the vapour pressure deficit (VPD) between the Temperature $T$ and humidity $H$, which is a non-linear function of temperature and humidity $f_{vpd}(T, H) = 0.6108 \times \exp\left(\frac{17.27 \times T}{T + 237.3}\right) \times (1 - H)$, where $T$ is the temperature in Celsius and $H$ is the relative humidity expressed as a decimal. This is a real-world example dataset that include the variables important for agriculture. The relational variable added to the Traffic (rTraffic) dataset California Department of Transportation (2024) is the sum of two other variables. The relational variable added to the Illness (rIllness) dataset Centers for Disease Control and Prevention (2024) is the difference

between two other variables. The relational variable added to the ETTH1 (rETT) dataset Zhou et al. (2021) is the product of two other variables. A context length of $L = 24$ is used for each data point, which is then flattened. A schematic of this for three variables is as shown in Fig. 1 (left top).

**Tasks**   We consider the following prompts, which will be referred to as tasks. Take the two variables and their corresponding relational variable and refer to them as $x_0$, $x_1$, and $x_2$, respectively. Let $\tau \in \{0, ..., L-1\}$ index the time-step within the data point (not to be confused with the diffusion time-step $t$). **Over-constrained (OC)**: $x_0, x_1$ and $\tau \in \{0, ..., L-1\}$ are used for the prompt. This task effectively test the model's ability to learn the deterministic relation $x_2 = f(x_0, x_1)$. **Under-constrained (UC)**: $x_2$ and $\tau \in \{0, ..., L-1\}$ are used for the prompt. This task effectively tests the model's ability to learn the probabilistic relation $x_3 \sim p(x_2|x_0, x_1)$. **Forecast (FC)**: All the variables and $\tau \in \{0, ..., L/2 - 1\}$ are used for the prompt. This also tests the model's capacity to learn a probabilistic function. Illustrations of the tasks are shown in the middle left of Fig. 1.

Table 1: Relational error $E_r$ for each model on each dataset (lower is better). The best values for each dataset are highlighted in bold. The mean values relative to the weak baseline are also given.

| DATASET | MODEL | TASK = OC | | TASK = UC | | TASK = FC | |
|---|---|---|---|---|---|---|---|
| | | $E_r$ | $\langle E_r \rangle / \langle E_r \rangle^{\text{(BASELINE)}}$ | $E_r$ | $\langle E_r \rangle / \langle E_r \rangle^{\text{(BASELINE)}}$ | $E_r$ | $\langle E_r \rangle / \langle E_r \rangle^{\text{(BASELINE)}}$ |
| rECL | BASELINE | $0.9841 \pm 0.3500$ | $1.0000$ | $0.9841 \pm 0.3500$ | $1.0000$ | $0.9841 \pm 0.3500$ | $1.0000$ |
| | DM (OURS) | $\mathbf{0.1491 \pm 0.0543}$ | $0.1515$ | $\mathbf{0.0572 \pm 0.0265}$ | $0.05812$ | $\mathbf{0.0138 \pm 0.0037}$ | $0.0140$ |
| | MOMENT | $0.5744 \pm 0.2019$ | $0.5837$ | $0.5495 \pm 0.2203$ | $0.5584$ | $0.2164 \pm 0.1272$ | $0.2199$ |
| | TIMER | $0.6197 \pm 0.2400$ | $0.6297$ | $0.5121 \pm 0.2127$ | $0.5203$ | $0.2182 \pm 0.1260$ | $0.2217$ |
| rWTH | BASELINE | $0.6283 \pm 0.3026$ | $1.0000$ | $0.6283 \pm 0.3026$ | $1.0000$ | $0.6283 \pm 0.3026$ | $1.0000$ |
| | DM (OURS) | $\mathbf{0.0550 \pm 0.0600}$ | $0.0875$ | $\mathbf{0.0932 \pm 0.0673}$ | $0.1483$ | $\mathbf{0.0160 \pm 0.0076}$ | $0.0255$ |
| | MOMENT | $0.2683 \pm 0.1820$ | $0.4270$ | $0.2651 \pm 0.1801$ | $0.4219$ | $0.0785 \pm 0.0507$ | $0.1249$ |
| | TIMER | $0.6477 \pm 0.2167$ | $1.0309$ | $0.3492 \pm 0.2179$ | $0.5558$ | $0.2459 \pm 0.0467$ | $0.3913$ |
| rTraffic | BASELINE | $0.1058 \pm 0.0579$ | $1.0000$ | $0.1058 \pm 0.0579$ | $1.0000$ | $0.1058 \pm 0.0579$ | $1.0000$ |
| | DM (OURS) | $\mathbf{0.0027 \pm 0.0010}$ | $0.0255$ | $\mathbf{0.0096 \pm 0.0056}$ | $0.0907$ | $\mathbf{0.0014 \pm 0.0006}$ | $0.0132$ |
| | MOMENT | $0.0513 \pm 0.0314$ | $0.4849$ | $0.0533 \pm 0.0326$ | $0.5038$ | $0.0046 \pm 0.0040$ | $0.0435$ |
| | TIMER | $0.0974 \pm 0.0284$ | $0.9206$ | $0.1006 \pm 0.0309$ | $0.9509$ | $0.0043 \pm 0.0035$ | $0.0409$ |
| rIllness | BASELINE | $4469 \pm 3585$ | $1.0000$ | $4469 \pm 3585$ | $1.0000$ | $4469 \pm 3585$ | $1.0000$ |
| | DM (OURS) | $\mathbf{1521 \pm 951.6}$ | $0.3403$ | $\mathbf{996.4 \pm 661.9}$ | $0.2230$ | $\mathbf{380.1 \pm 224.7}$ | $0.0851$ |
| | MOMENT | $3183 \pm 1913$ | $0.7122$ | $3815 \pm 2098$ | $0.8537$ | $1174 \pm 681.0$ | $0.2627$ |
| | TIMER | $3314 \pm 1545$ | $0.7416$ | $3459 \pm 2096$ | $0.7740$ | $1554 \pm 1384$ | $0.3477$ |
| rETT | BASELINE | $0.5600 \pm 0.2894$ | $1.0000$ | $0.5600 \pm 0.2894$ | $1.0000$ | $0.5600 \pm 0.2894$ | $1.0000$ |
| | DM (OURS) | $\mathbf{0.2312 \pm 0.1704}$ | $0.4129$ | $\mathbf{0.2875 \pm 0.1750}$ | $0.5134$ | $\mathbf{0.0597 \pm 0.0398}$ | $0.1066$ |
| | MOMENT | $\mathbf{0.3796 \pm 0.2392}$ | $0.6779$ | $\mathbf{0.3231 \pm 0.1977}$ | $0.5770$ | $0.1440 \pm 0.0908$ | $0.2571$ |
| | TIMER | $\mathbf{0.3177 \pm 0.2185}$ | $0.5673$ | $0.4666 \pm 0.2721$ | $0.8332$ | $0.2271 \pm 0.1324$ | $0.4055$ |

**Models**   that will be evaluated on the tasks above are:

- **Baseline** - Since each dataset will have different scales, a baseline is required to compare against. A weak baseline that returns the training set mean for each variable for all responses will be used.

- **Diffusion Model** - The diffusion model trained on each dataset, which will be used for hallucination detection on that dataset. It can also be used for question answering. This will serve as a stronger baseline. The model uses a simple five layer MLP ~1M parameters.

- **MOMENT** Goswami et al. (2024) - A MVTS FM using a transformer encoder architecture with 24 layers and 385M parameters, pre-trained on Time-Series Pile (20GB). This model will be used for question answering only. MOMENT models MVTS in a channel-independent manner, a popular choice Nie et al. (2022). As shown in Fig. 1 (left bottom), we therefore provide additional context (24 time-steps) to each task to allow MOMENT to function on tasks like the OC and UC task. This makes the task easier.

- **TIMER** Liu et al. (2024) - A MVTS FM using a transformer decoder architecture with 4 layers and 2M parameters, pre-trained on the UTSD-4G dataset (1.2GB). This model will be used for question

answering only. Since TIMER requires at least the first token (24 time-steps) to be provided, additional context is also provided in the same way as MOMENT. This allows for a fair comparison.

**Implementation** is in Python 3.11 using PyTorch. The diffusion models trained were all MLPs with five hidden layers of size 512. A linear variance schedule ranging from a value of 1e-4 to 1e-2 was used with 1000 diffusion steps. Models were trained using the ADAM optimizer Diederik (2014), one-cycle learning rate scheduler Smith & Topin (2019), a maximum learning rate of 1e-3 and batch size of 1024. All models were trained up to a maximum of 8000 epochs with early stopping. The model with best validation loss was used for all subsequent experiments. The relational datasets use all the data present in the original dataset and were split into train, validation and test sets with a ratio of 5:1:1 in a chronologically increasing manner such that there is no overlap in time. Training runs on a single NVIDIA T1000 in 2-22 hours depending on the dataset.

## 4.1 Multi-Variate Time-Series Models Hallucinate

Using our evaluation method described above, the degree of relational hallucination exhibited by a model can be quantified. This is achieved by using each model to respond to all the prompts from the OC, UC and FC tasks on each dataset (test set), and then computing the relational error $E_r$. The lower the average $E_r$ is, the better. As each dataset has different value scales, all $E_r$ comparisons are relative to the weak baseline. Since the diffusion model was trained on the training set of each dataset, it can be taken as a strong baseline. These values are shown in Tab. 1 (mean and standard deviation) for each model, task and dataset (test set). The mean values normalised by the baseline's mean is also provided so that it is easier to compare across the datasets with different scales.

The results show that even with the handicap of being given extra context, both the pre-trained FMs (MOMENT and TIMER) hallucinate heavily. They typically hallucinate less than the weak baseline but in some cases can match or even exceed it. The diffusion model (strong baseline) hallucinates the least, but nevertheless still hallucinates. All models relationally hallucinate the least on the FC task. This may be because there are no hard constraints on the values that must be predicted and the model is free to sample/predict values that are relationally correct. Averaging over the tasks and datasets, the relational hallucination level of the diffusion model, MOMENT and TIMER are 15.3%, 44.6% and 59.5% the values of the weak baseline. The results demonstrate that even models trained on each dataset can relationally hallucinate relative to that dataset, with this being exhibited much strongly in pre-trained FMs.

## 4.2 Estimation of Hallucination Levels at Inference

The following proposed quartile thresholding method is used to classify responses by their expected relational hallucination level: low, medium and high. This can be evaluated by computing the relational error $E_r$ for all the prompt-response pairs classified into each class. This gives us the distribution of $E_r$ for each class. The overlap coefficient between the distribution of $E_r$ for the low and high classes can be computed. These distributions will be referred to as $\mathbf{P}^{(\mathrm{L})}$ and $\mathbf{P}^{(\mathrm{H})}$, respectively. They are of the form $\mathbf{P} = \{P_0, P_1, ..., P_{n-1}\}$, where $n$ is the number of bins and the values are the probability in each bin with $\sum_k P_k = 1$. The overlap coefficient between them can be computed as $\sum_{k=0}^{n-1} \min\left(P_k^{(\mathrm{L})}, P_k^{(\mathrm{H})}\right)$. Lower coefficients mean better hallucination detection. A value of zero implies zero overlap, and a value of one implies that the distributions are identical. The results (mean and standard deviation) obtained for the models on each dataset averaged over five runs are shown in Tab. 2. The overlap coefficients are low (generally below 1%) except for the the rETT dataset which is a moderate value of around 15%. The histogram of the relational error for each class of hallucination level is shown in Fig. 2. The results show that quartile thresholding is a simple and effective way to classify responses into their expected relational hallucination levels where the distributions with high and low hallucination have low overlap.

Table 2: Overlap coefficient between the data distribution classified as low and high hallucination (lower is better). Relative change in relational error $\Delta_{E_r}$ of the selected response using filtering (lower is better).

| DATASET | MODEL | OVERLAP COEFFICIENT | $\Delta_{E_r}$(OC) | $\Delta_{E_r}$ (UC) | $\Delta_{E_r}$ (FC) |
|---|---|---|---|---|---|
| rECL | DM (OURS) | $0.0008 \pm 0.0003$ | $0.6230 \pm 0.1060$ | $0.4789 \pm 0.1045$ | $0.6705 \pm 0.1061$ |
| | MOMENT | $0.0000 \pm 0.0000$ | $0.7397 \pm 0.1159$ | $0.7549 \pm 0.12$ | $0.7249 \pm 0.2889$ |
| | TIMER | $0.0000 \pm 0.0000$ | $0.7289 \pm 0.1854$ | $0.705 \pm 0.1919$ | $0.5468 \pm 0.2581$ |
| rWTH | DM (OURS) | $0.0167 \pm 0.0007$ | $0.7051 \pm 0.1656$ | $0.5089 \pm 0.1782$ | $0.8550 \pm 0.2287$ |
| | MOMENT | $0.0005 \pm 0.0005$ | $0.8086 \pm 0.1195$ | $0.8143 \pm 0.1242$ | $0.7231 \pm 0.239$ |
| | TIMER | $0.0002 \pm 0.0001$ | $0.8175 \pm 0.149$ | $0.7877 \pm 0.1752$ | $0.7753 \pm 0.1772$ |
| rTRAFFIC | DM (OURS) | $0.0009 \pm 0.0003$ | $0.6600 \pm 0.0902$ | $0.4493 \pm 0.1162$ | $0.8057 \pm 0.2055$ |
| | MOMENT | $0.0000 \pm 0.0000$ | $0.7769 \pm 0.1229$ | $0.7638 \pm 0.1197$ | $0.8739 \pm 0.1409$ |
| | TIMER | $0.0000 \pm 0.0000$ | $0.7743 \pm 0.1679$ | $0.7985 \pm 0.161$ | $0.5228 \pm 0.2608$ |
| rILLNESS | DM (OURS) | $0.0111 \pm 0.0040$ | $0.7311 \pm 0.1372$ | $0.7079 \pm 0.1225$ | $0.8787 \pm 0.1986$ |
| | MOMENT | $0.0008 \pm 0.0017$ | $0.8392 \pm 0.2418$ | $0.7860 \pm 0.1810$ | $0.9647 \pm 0.3061$ |
| | TIMER | $0.0000 \pm 0.0000$ | $0.6475 \pm 0.2664$ | $0.5988 \pm 0.2501$ | $0.5343 \pm 0.3847$ |
| rETT | DM (OURS) | $0.1538 \pm 0.0054$ | $0.7681 \pm 0.2128$ | $0.6724 \pm 0.2382$ | $0.9074 \pm 0.3903$ |
| | MOMENT | $0.0880 \pm 0.0045$ | $0.8291 \pm 0.1645$ | $0.8269 \pm 0.1597$ | $0.7353 \pm 0.3114$ |
| | TIMER | $0.0371 \pm 0.0061$ | $0.7752 \pm 0.2439$ | $0.7833 \pm 0.1976$ | $0.672 \pm 0.2444$ |

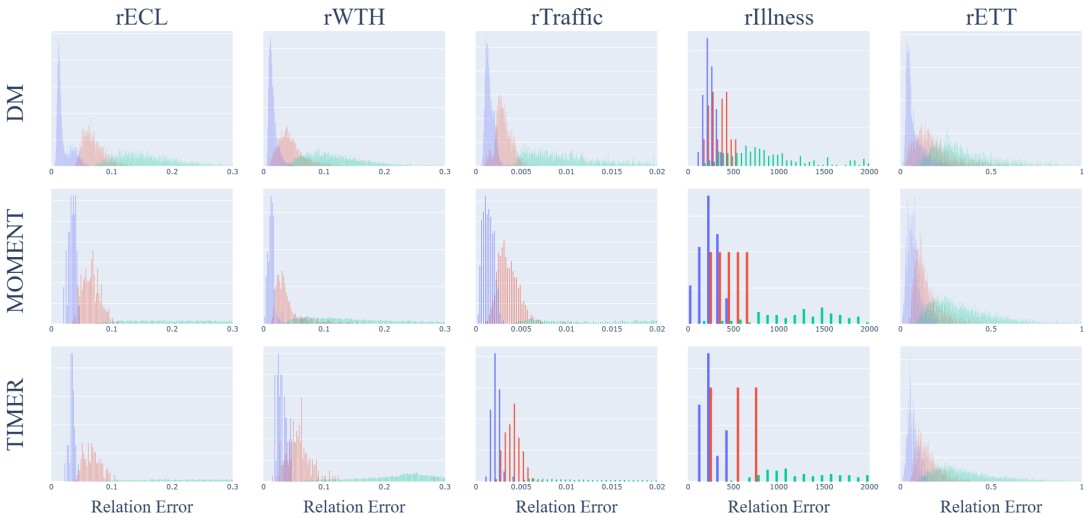

Figure 2: Histogram showing the distribution of relational error for the data points with expected low (blue), medium (red) and high (green) hallucination level. The $x$-axis is the relational error and the $y$-axis is the probability. The subplots are aligned by dataset (column) and model (row).

## 4.3 Mitigation of Hallucination at Inference

The proposed filtering method for mitigating relational hallucination can be evaluated by computing the relational error $E_r$ for the response selected by filtering $E_r^{(j^*)}$ and comparing it to the mean relational error for the $N$ sampled responses $\langle E_r \rangle = \frac{1}{N} \sum_{j=1}^{N} E_r^{(j)}$. The relative change in relational error can then be computed $\Delta_{E_r} = E_r^{(j^*)}/\langle E_r \rangle$. The lower $\Delta_{E_r}$ is the better, with $\Delta_{E_r} = 1$ meaning there is no improvement. Since the FMs (MOMENT and TIMER) are deterministic, a simple way to make sample from them is to

activate the dropout layers used for their training. The sample with the lowest CE is then selected. Instead of computing $\Delta_{E_r}$ relative to the mean of the ensemble, it should be relative to the response from the model with deactivated dropout.

The relative change in relational error $\Delta_{E_r}$ for each dataset averaged over 20 runs is given in Tab. 2 (mean and standard deviation). The average relative change $\Delta_{E_r}$ is always less than unity, which means that the filtering method is effective, even when the pre-trained FMs with dropout. The proposed method can on average reduce the relational error by up to 55.0% for the diffusion model and 47.7% for the pre-trained FMs. This demonstrates that filtering using CE is a simple and effective method for mitigating relational hallucination.

## 5 Conclusion

Hallucination in MVTS imputation has been defined using analogies from established definitions in NLP. Pre-trained open-source MVTS FMs are seen to hallucinate in this manner on our datasets. By training a diffusion model on data in a target domain and extracting the proposed CE metric, it is possible to detect and mitigate MVTS hallucination, being able to on average reduce the hallucination of pre-trained FMs by up to 47.7%. This work encourages the responsible use of MVTS FMs by formally defining, detecting and mitigation MVTS hallucination.

**Limitations and Further Work**  While our work shows promising results, it is largely empirical. For instance, our mitigation method statistically improves responses, but is not guaranteed to always do so. Additionally, the MLP architecture used for the diffusion model is simple, and hence does not naturally support variable length responses. Since we stack each variable into one time-series as the input to the model, the simple MLP architecture does not scale well to a high number of variables or long windows. This is however not a limitation with the method itself but rather a design choice chosen for simplicity. Future work can be done to investigate neural architecture choices and the use of latent diffusion or tokenisation. Although the current method used to convert deterministic pre-trained MVTS FMs into non-determinstic ones that can be sampled works, it is very simple. Exploring decoding strategies and methods from NLP for sampling responses that can be applied to MVTS is another promising direction.

## 6 Reproducibility Statement

We provide the necessary details to ensure the reproducibility of our work. The theoretical preliminaries required for our methods are provided in Section 1.1. Our proposed method and approach are described in Section 3. Implementation details, including hardware and software, training procedures, experimental setting, data processing and information on models are presented in Section 4. Sources and licenses for the standard datasets and pre-trained models used in our work are provided in Appendix E. Additional information on training procedures, experimental settings and data processing is detailed in the provided source code, which contains instructions as part of a README file.

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

## A   Other Metrics

It has been shown on a computer vision and toy Gaussian dataset that a measure of hallucination can be extracted from unconditional diffusion models during the generation process Aithal et al. (2024). This measure will be referred to as the the trajectory variance (TV), which is the variance of the predicted mean with respect to the diffusion time-step. The predicted mean at each diffusion time-step (Eq. 4) is obtained from the generation process and will be written as $\mu_{i,t}$, where $i$ indexes the data dimension and $t$ indexes the diffusion time-step. The TV metric is calculated as

$$M_{\text{TV}} = \text{Mean}_i\bigg(\text{Var}_t(\mu_{i,t})\bigg), \tag{10}$$

where variance is taken across the diffusion time-step and mean across the data dimension. A schematic example of this is shown in Fig. 3. This measures the variation in the trajectory of the variables during the diffusion process.

TV however only applies to unconditional generation and does not apply to the prompt-response framework using imputation. This is because in the prompt-response framework, the subset of the data dimension that is used for the prompt is not unconditionally generated. Three modifications to the TV metric that address this are proposed. These are response trajectory spread (RTS), prompt trajectory spread (PTS) and combined trajectory spread (CTS) metrics. We also propose two additional metrics that use the magnitude of the noise returned by the diffusion model as a metric to detect hallucination. These are prompt error (PE) and combined error (CE). The combined error is the metric presented in the main text as this is the most effective metric and the other metrics fail at detecting relational hallucination.

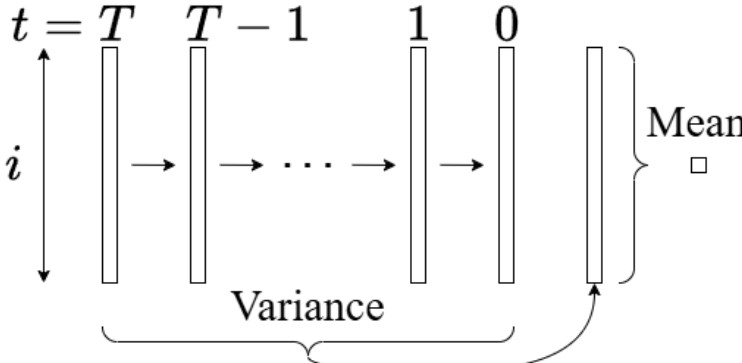

Figure 3: Schematic showing the computation of the trajectory variance (TV) metric. The variance is taken across the diffusion time-step $t$ and the mean is taken across the data dimension $i$.

**Response Trajectory Spread (RTS)**  Since TV is computed for unconditional generation, the simplest generalisation to the prompt-response framework is to compute this metric for the response only, as this is the part which is generated in a similar manner. Instead of using the variance however, this work uses the standard deviation since it is simpler and is more interpretable. The response trajectory spread (RTS) can be computed as

$$M_{\mathrm{RTS}} = \mathrm{Mean}_i\Big(\mathrm{Std}_t(\mu_{i,t})\Big), \ \forall i \in \mathcal{I}_{\mathrm{r}}, \tag{11}$$

where standard deviation is taken across the diffusion time-step, and mean across the data dimension. This is illustrated in Fig. 3 but with standard deviation instead of variance.

**Prompt Trajectory Spread (PTS)**  As we are using RePaint Lugmayr et al. (2022) to condition the diffusion model, all predicted means of the prompt are clamped to the values provided by the prompt. The values provided by the prompt can therefore be used as the mean that is required to compute the standard deviation. The prompt trajectory spread (PTS) can be computed as

$$M_{\mathrm{PTS}} = \mathrm{Mean}_i\Big(\mathrm{RMSE}_t(\mu_{i,t}, x_i)\Big), \ \forall i \in \mathcal{I}_{\mathrm{p}}, \tag{12}$$

where the root mean square error is taken across the diffusion time-step and the mean across the data dimension.

**Combined Trajectory Spread (CTS)**  The final output from the model $\hat{x}_i$ which combines both the prompt and the response can also be used to compute the trajectory spread. The full diffusion process can be computed one more time by setting the prompt as $x_i = \hat{x}_i, \forall i \in \mathcal{I}$. The combined trajectory spread (CTS) can then be computed for this as

$$M_{\mathrm{CTS}} = \mathrm{Mean}_i\Big(\mathrm{RMSE}_t(\mu_{i,t}, \hat{x}_i)\Big), \ \forall i \in \mathcal{I}, \tag{13}$$

where the root mean square error $RMSE_t(\mu_{i,t}, x_i) = \sqrt{\frac{1}{T}\sum_{t=1}^{T}(\mu_{i,t} - x_i)^2}$ is taken across the diffusion time-step and the mean is taken across the data dimension. Since the full diffusion process has to be computed completely an additional time, this metric is computationally expensive. CTS is like PTS but includes both the prompt and response.

**Prompt Error (PE)**  The previous metrics are all based on the trajectory variance Aithal et al. (2024). This work proposes two additional simple metrics based on the reconstruction error of the final output of the

model. The first considers the reconstruction error of the output with respect to the prompt. Only indices $i \in \mathcal{I}_\mathrm{P}$ are used since as are the only values where ground-truth is available through the values provided by the prompt. PE can be computed as

$$M_\mathrm{PE} = \mathrm{RMSE}_i(\hat{x}_i, x_i), \ \forall i \in \mathcal{I}_\mathrm{p}, \tag{14}$$

where the RMSE is taken across the data dimension.

**Combined Error (CE)** The PE metric can be extended to also include the response indices $\mathcal{I}_\mathrm{R}$ in the same way as the CTS metric, which leads to the CE metric of Eqn. 9.

## A.1 Sensitivity to Relational Hallucination

**RTS** The sensitivity of the RTS metric to the relational error on the test set for each task and dataset is shown in Fig. 4. The metric is not sensitive to the relational error.

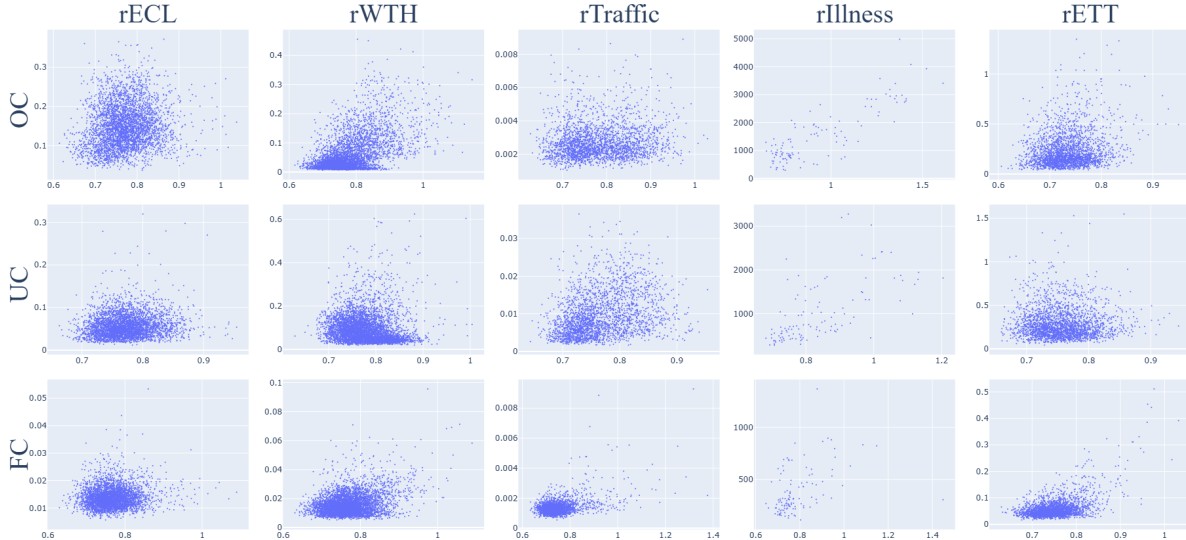

Figure 4: Scatter plot showing the relationship between the RTS metric ($x$-axis) and the ground-truth relational error ($y$-axis) on the test set. The subplots are aligned by dataset (column) and task (row). The axis limits are the same within each dataset (column).

**PTS** The sensitivity of the PTS metric to the relational error on the test set for each task and dataset is shown in Fig. 5. The metric is not sensitive to the relational error.

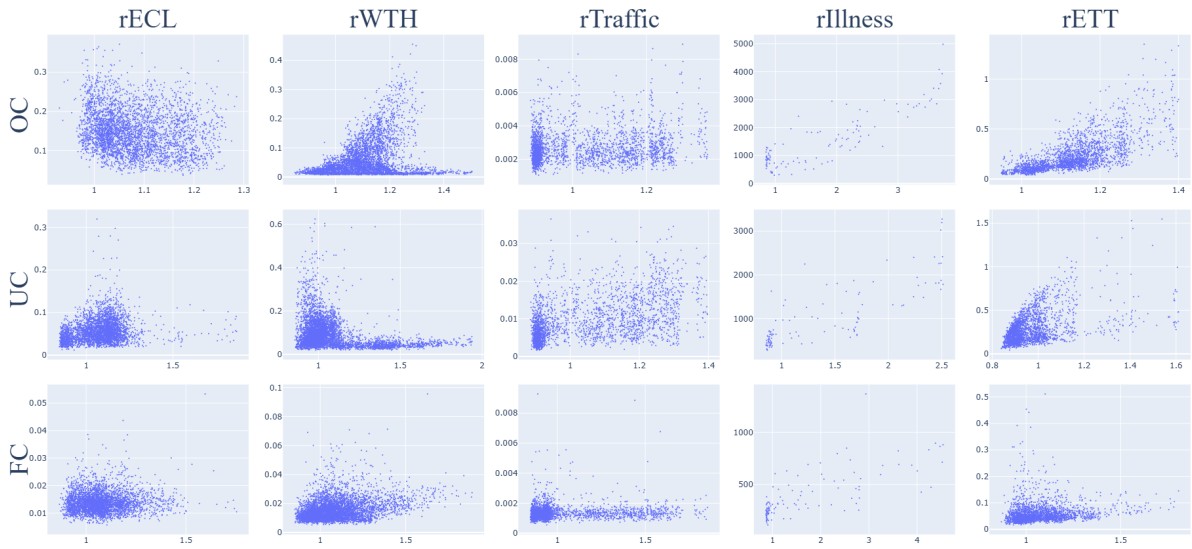

Figure 5: Scatter plot showing the relationship between the PTS metric ($x$-axis) and the ground-truth relational error ($y$-axis) on the test set. The subplots are aligned by dataset (column) and task (row). The axis limits are the same within each dataset (column).

**CTS** The sensitivity of the CTS metric to the relational error on the test set for each task and dataset is shown in Fig. 6. The metric is not sensitive to the relational error.

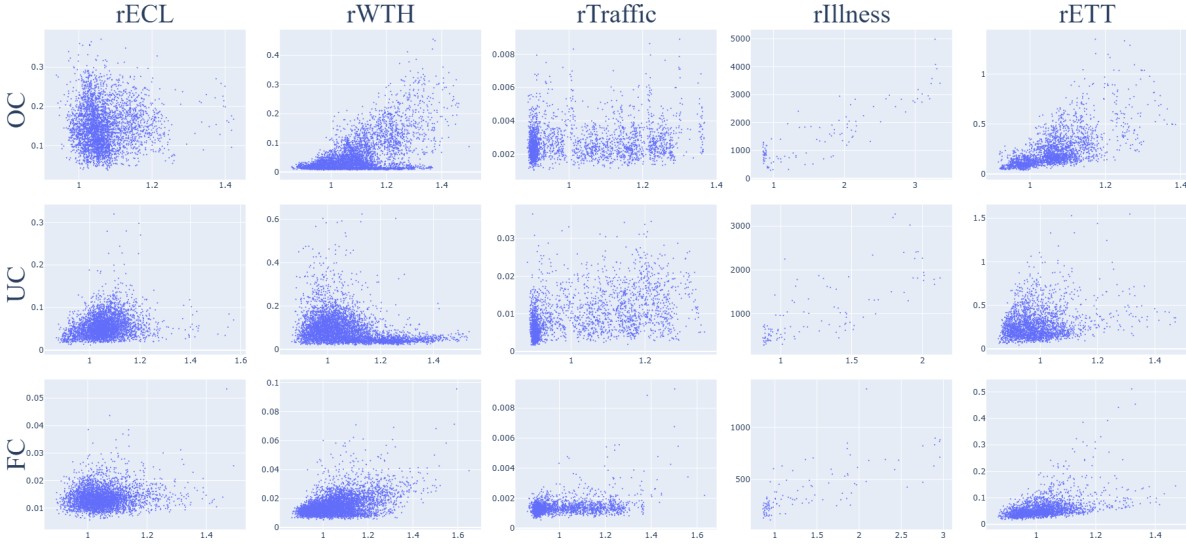

Figure 6: Scatter plot showing the relationship between the CTS metric ($x$-axis) and the ground-truth relational error ($y$-axis) on the test set. The subplots are aligned by dataset (column) and task (row). The axis limits are the same within each dataset (column).

**PE** Sensitivity of PE to relational error on the test set for each task and dataset is shown in Fig. 7. PE is not as robustly and consistently sensitive to the relational error as the CE metric, which is shown in Fig. 8. This may be because PE only includes the prompt, and since relational hallucination is the inconsistency of a prompt-response pair, it is expected that a metric including both the prompt and response such as CE would perform better.

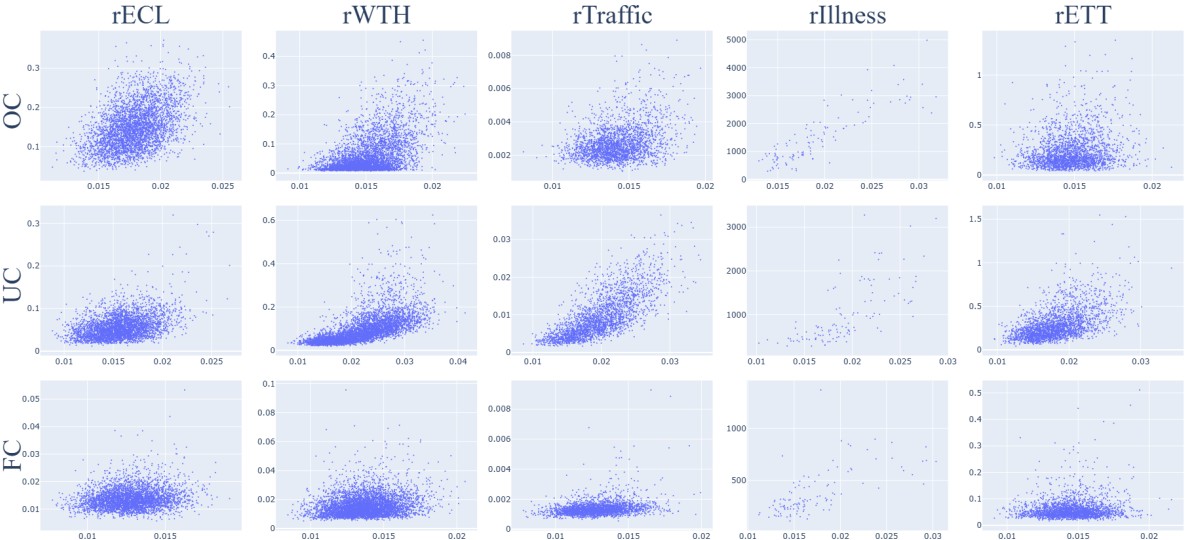

Figure 7: Scatter plot showing the relationship between the PE metric ($x$-axis) and the ground-truth relational error ($y$-axis) on the test set. The subplots are aligned by dataset (column) and task (row). The axis limits are the same within each dataset (column).

**CE** Sensitivity of CE to relational error on the test set for each task and dataset is shown in Fig. 8. The CE metric is sensitive to the relational error.

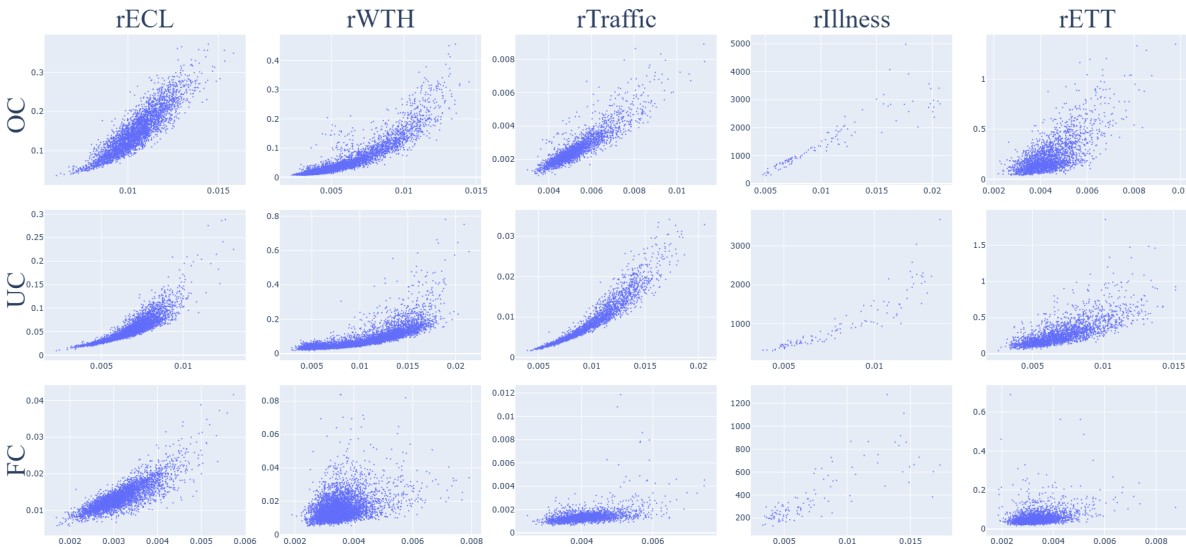

Figure 8: Scatter plot showing the relationship between the CE metric ($x$-axis) and the ground-truth relational error ($y$-axis) on the test set. The subplots are aligned by dataset (column) and task (row). The axis limits are the same within each dataset (column).

# B    Out of Distribution Behavior

The difference in behavior between distributional hallucination (OOD) and relational hallucination can be studied by probing the diffusion model with prompts that are on the edge of the training distribution in the data space. One can achieve this by taking prompts from the training set and pushing it out of distribution in some way. Two ways to achieve this is to apply a constant offset to the prompt (this preserves the prompt shape but pushes the values out of distribution), or to flatten the prompt to the mean of that prompt (this pushes the prompt shape out of distribution but leaves the values in-distribution). As shown in Figures 9 and 10, respectively, the relationship between CE metric and the ground truth $E_r$ under these conditions is maintained. Particularly, this relationship holds as the offset increases. This shows that prompts that are most out of distribution but are relationally correct can still be detected using the CE metric.

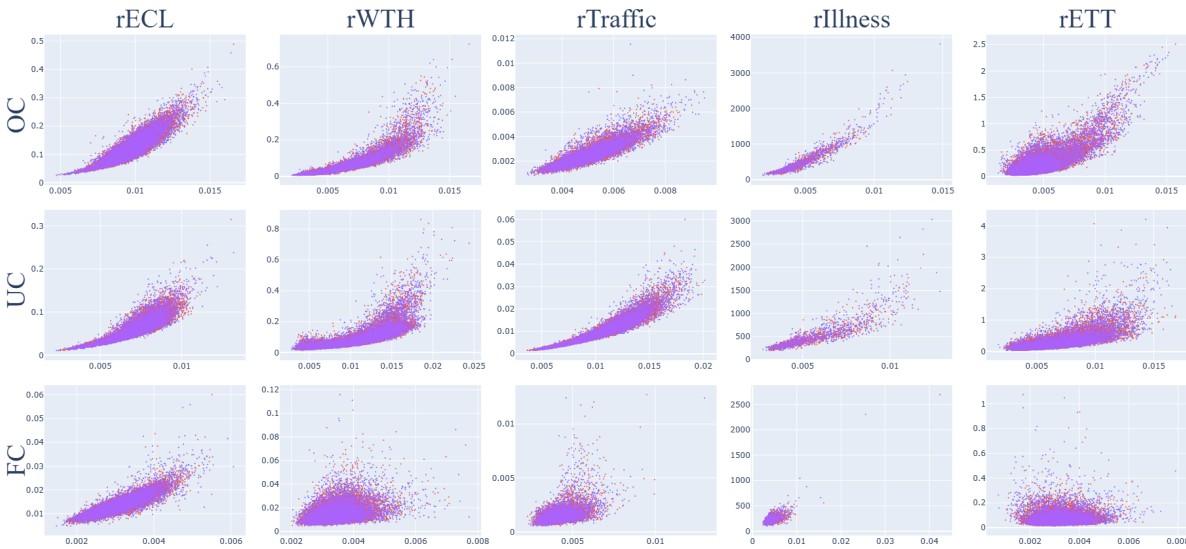

Figure 9: Scatter plot showing the relationship between the CE metric ($x$-axis) and the ground-truth relational error ($y$-axis) for out of distribution data constructed by offsetting in-distribution prompts. Points with blue, red and purple colors correspond to points that are increasingly out of distribution, respectively. The subplots are aligned by dataset (column) and task (row). The axis limits are the same within each dataset (column).

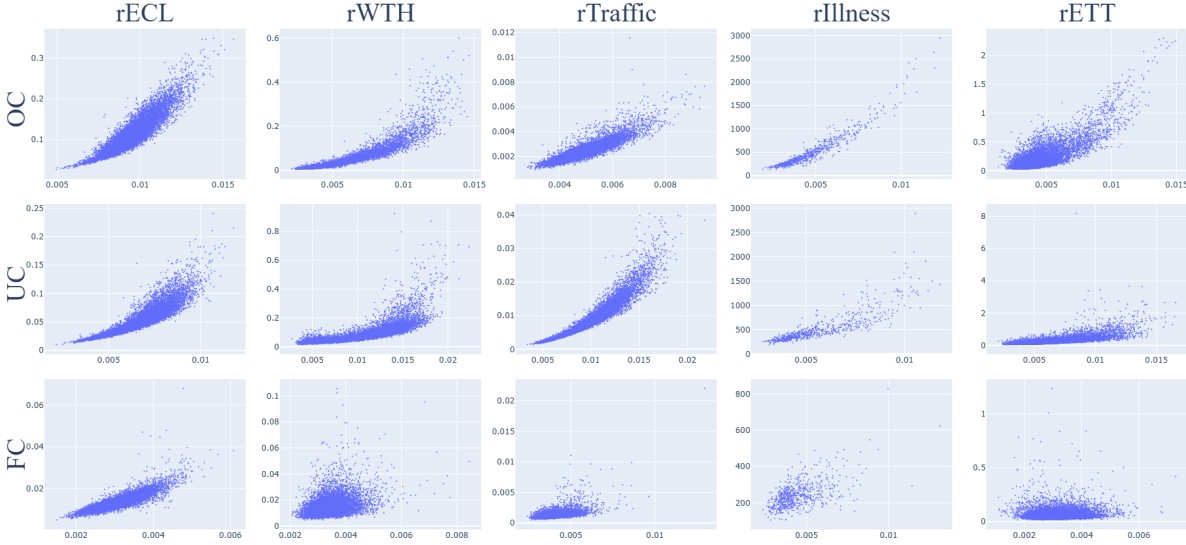

Figure 10: Scatter plot showing the relationship between the CE metric ($x$-axis) and the ground-truth relational error ($y$-axis) for out of distribution data constructed by flattening variables. The subplots are aligned by dataset (column) and task (row). The axis limits are the same within each dataset (column).

## C    Filtering

Our proposed filtering method mitigates hallucination by sampling $N$ responses and selecting the response with the lowest CE. The reduction in hallucination levels $\Delta_{E_r}$ as $N$ is increased is shown in Figure 11 for all the tasks, datasets and models. As expected, $\Delta_{E_r}$ decreases as $N$ is increased, with the 'elbow' of the plots occurring around the value of $N = 20$. The value of $N = 20$ is what is used in the main body of this work.

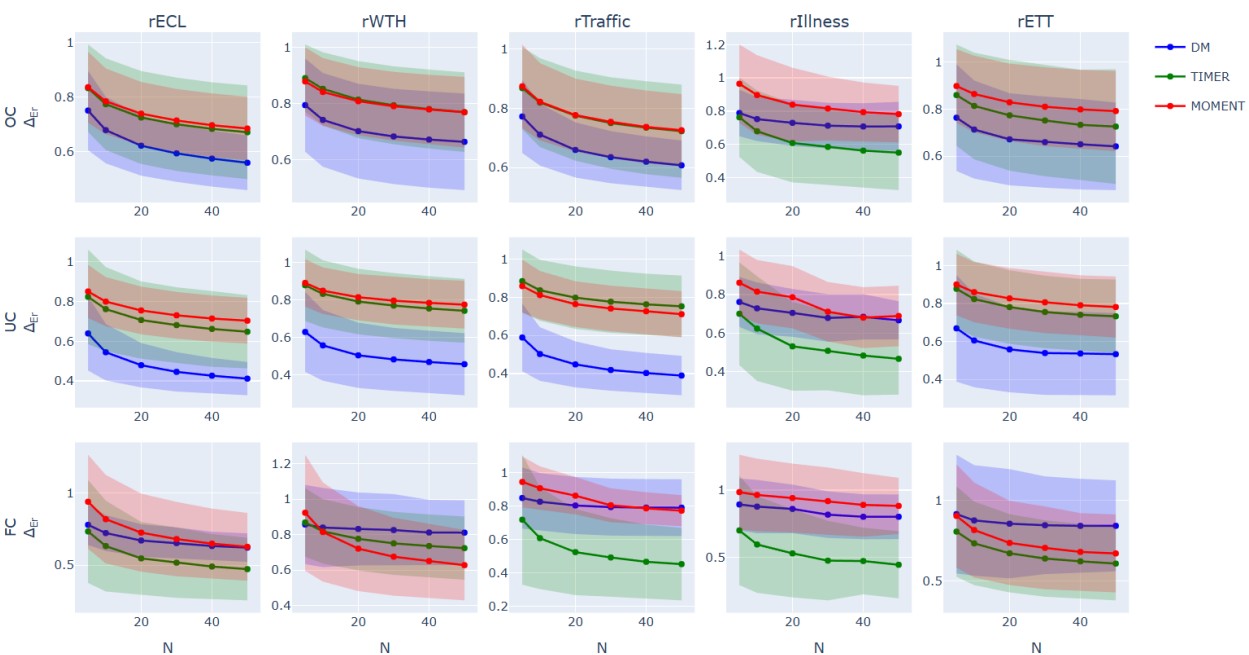

Figure 11: The reduction in hallucination levels $\Delta_{E_r}$ as the number of samples $N$ is increased.

## D    Change in Imputation Error After Hallucination Mitigation

The relative change in imputation error $\Delta E_i$ after applying the proposed filtering method to mitigates hallucination is shown in Tab. 3. This is calculated in the same way as described in Section 4.3 but with the imputation error $E_i$ instead. A value of 1.0 means that there is no change in imputation error. A value $< 1.0$ implies that the imputation error is lower than the pre-mitigated version (which is better). The results show that the imputation error remains statistically unchanged after applying the proposed hallucination mitigation method, or, in some cases it is improved.

## E    Licenses for Existing Assets

### E.1    Datasets

The datasets are commonly used MVTS time-series datasets and can be accessed from the Autoformer repository (https://github.com/thuml/Autoformer) which is under MIT license.

- ECL - CC BY 4.0

- WTH - N/A

- Traffic - CC BY 4.0

- Illness - N/A

Table 3: Relative change in imputation error $\Delta_{E_r}$ of the selected response using filtering method (lower is better, a value of 1.0 implies no change).

| DATASET | MODEL | $\Delta_{E_i}$ (OC) | $\Delta_{E_i}$ (UC) | $\Delta_{E_i}$ (FC) |
|---|---|---|---|---|
| rECL | DM (OURS) | $0.9795 \pm 0.1091$ | $0.9282 \pm 0.1706$ | $0.9684 \pm 0.1556$ |
| | MOMENT | $1.0170 \pm 0.0566$ | $1.0040 \pm 0.0448$ | $1.0140 \pm 0.0606$ |
| | TIMER | $0.9917 \pm 0.0888$ | $0.9969 \pm 0.0667$ | $0.9988 \pm 0.0663$ |
| rWTH | DM (OURS) | $0.9792 \pm 0.1152$ | $0.8425 \pm 0.1352$ | $0.9171 \pm 0.1919$ |
| | MOMENT | $1.0870 \pm 0.1344$ | $0.9933 \pm 0.0544$ | $1.0170 \pm 0.0880$ |
| | TIMER | $1.0190 \pm 0.1768$ | $0.9949 \pm 0.1167$ | $0.9850 \pm 0.0828$ |
| rTRAFFIC | DM (OURS) | $0.8275 \pm 0.0981$ | $0.8977 \pm 0.1059$ | $0.8732 \pm 0.1451$ |
| | MOMENT | $0.9933 \pm 0.0599$ | $1.0280 \pm 0.0651$ | $1.0270 \pm 0.0739$ |
| | TIMER | $0.9437 \pm 0.1067$ | $0.9868 \pm 0.0988$ | $1.0030 \pm 0.0975$ |
| rILLNESS | DM (OURS) | $0.8058 \pm 0.1245$ | $0.9211 \pm 0.1447$ | $0.9295 \pm 0.1097$ |
| | MOMENT | $1.0700 \pm 0.0782$ | $0.9786 \pm 0.0582$ | $0.9991 \pm 0.0668$ |
| | TIMER | $0.9501 \pm 0.1304$ | $0.9653 \pm 0.0929$ | $0.9664 \pm 0.1040$ |
| rETT | DM (OURS) | $0.8724 \pm 0.1109$ | $0.8568 \pm 0.1384$ | $0.9534 \pm 0.1846$ |
| | MOMENT | $1.0400 \pm 0.0986$ | $1.0030 \pm 0.0703$ | $1.0350 \pm 0.1123$ |
| | TIMER | $0.9962 \pm 0.1665$ | $0.9887 \pm 0.0711$ | $0.9903 \pm 0.0807$ |

- ETT - CC BY-ND 4.0

### E.2 Models

- MOMENT - MIT (https://github.com/moment-timeseries-foundation-model/moment)

- TIMER - MIT (https://github.com/thuml/Large-Time-Series-Model)

## F   Use of Large Language Models

Large language models were not used beyond grammar checking and polishing writing.

