# OpenReview forum: "Hallucination Detection and Mitigation with Diffusion in Multi-Variate Time-Series Foundation Models"
_TMLR — Accepted by TMLR_

### Review · Reviewer_f54C · 2026-04-06

**Summary Of Contributions:**

This paper attempts to translate NLP hallucination concepts into multi-variate time-series (MVTS) by defining "distributional" and "relational" hallucinations for imputation tasks. It introduces a "Combined Error" (CE) metric, which uses a small, target-trained diffusion model to measure the consistency of a Foundation Model's output via a single denoising step. It also proposes a filtering heuristic to mitigate errors by selecting sampled responses with the lowest CE score.

Strengths:
The paper explores a relevant problem on the reliability of emerging MVTS Foundation Models and offers a computationally lightweight heuristic for evaluating output consistency.

Weaknesses:
The contributions lack theoretical rigor and risk merely rebranding standard Out-of-Distribution (OOD) or anomaly detection as "hallucination." The methodology is overly simplistic; the "diffusion model" used is merely a 5-layer MLP, which cannot naturally scale to complex or variable-length time series. Writing is also not ready for an academic paper.

**Audience:**

No

**Audience Explanation:**

Need to re-think if the core motivation is solid and whether the claims and results are consistent.

**Claims And Evidence:**

No

**Claims Explanation:**

**1. Artificial and Synthetic Evaluation Framework**
The definition of relational error $E\_r$ relies on an exact mathematical relation $f$. Because the true underlying mathematical formulas of real-world datasets are generally unknown or highly complex, the authors resort to making up synthetic variables (e.g., simply adding two columns together to make a third). Consequently, the entire experimental validation only proves that the proposed method can detect when a model fails basic arithmetic across channels. It does not demonstrate whether the method can detect organic, complex, or latent temporal hallucinations in genuine real-world data.

**2. Absence of Standard Task Performance Metrics**
The paper evaluates the models strictly on the synthetic "relational error" ($E_r$) and completely omits standard prediction/imputation performance metrics such as MSE or MAE. Because the MSE/MAE of the imputed values against the actual ground truth is not reported, there is no way to know if MOMENT and TIMER are actually successfully imputing the time-series or simply generating complete noise. We only know whether their generated variables add up correctly according to the injected synthetic math formula.

It is entirely possible that the sample with the best internal consistency (lowest CE) is actually the *least accurate* prediction compared to the true missing data. By omitting standard task performance, the paper fails to prove that this hallucination mitigation does not actively degrade the model's actual utility.

**3. Architectural Simplicity vs. Foundation Models**
The diffusion model used for hallucination detection is merely a 5-layer MLP, which is incredibly simple and small compared to the chosen Foundation Models. The paper does not address how this simple architecture performs when scaled up or when handling longer, variable-length sequences, limiting its general applicability.

**4. Questionable Premise of Time-Series FMs**
As shown in Table 1, the extremely small, target-trained diffusion model heavily outperforms the massive Foundation Models on these tasks. If a simple, localized model performs much better than common FMs, it indicates that large models with massive parameter counts do not necessarily carry more actionable knowledge for multi-variate time-series tasks. This raises a fundamental question: what is the actual use case for huge Foundation Models in this domain? The authors take for granted that the success of language modeling can be naturally extended to the time-series domain, but their own baseline results suggest otherwise.

**5. Writing Quality**
The writing is not ready for an academic paper. See more below.

**Requested Changes:**

### Writing Errors

**1. Overloaded and Confusing Notation:**

In Section 3 (Notations), the authors write: *"The final output (prediction) from the model (t = 0) will be denoted as $\hat{x}_i$, where the imputed response is $\hat{x}_i, \forall i \in \mathcal{I}_r$, and $\hat{x}_i \approx x_i, \forall i \in \mathcal{I}_p$."*
The variable $\hat{x}_i$ is used simultaneously to denote the entire output, the subset representing the imputed response, and the subset representing the prompt. Overloading the exact same notation is mathematically confusing.

**2. Awkward Typographical Choices:**
The use of a "double hat" ($\hat{\hat{x}}_i$) in Equation 9 and the surrounding text to represent the diffusion model's reconstruction is visually awkward and non-standard. Using a distinct variable identifier (such as $\tilde{x}_i$ or $x'_i$) would greatly improve readability.

**3. Unclear Table Formatting:**
In Table 1, multiple rows of results are bolded within the same column. It makes it difficult to parse which model actually performs best.

**4. Mathematical Typos:**
At the bottom of page 6, when describing the Under-constrained (UC) task, the text states the task tests the model's ability to learn the relation: $x_3 \sim p(x_2|x_0, x_1)$. However, $x_3$ is never defined in this context—only variables $x_0, x_1,$ and $x_2$ were established.

---

> ### Author Response · Authors · 2026-05-04
>
> Weaknesses:
> 1. We explained in the 'Related Concepts' paragraph in Section 2, how the proposed definition of relational hallucination are related to and differ from concepts such as OOD or anomaly detection. We demonstrate that pre-trained foundation models hallucinate and propose a simple method for detecting and for mitigating it, which we do not see as a downside.
>
> No convincing evidence:
> 1. We indeed add synthetic variables, which include both simple arithmetic relations but also complex non-linear ones as shown by the VPD equation in the ‘Relational Variables’ paragraph in Section 4. We also note that the VPD case is an example of real-world data which is useful for agriculture.
>
> It is important to note that even for the case of simple arithmetic relations, these result in complex non-linear relations that the model has to learn, which is discussed extensively in the ‘Tasks’ paragraph in Section 4. Relationships of the form $x_3 = f(x_1, x_2)$ may seem simple, since you are aware of the structure, but the models do not know of this structure and cannot exploit it's simplicity - they have to model the full joint distribution $p(x_1, x_2, x_3)$.
>
> It is also important to note that the OC, FC and UC tasks were designed explicitly to test for cases where the relations are complex, cannot be explicitly written down and are even probabilistic.  Indeed the OC task is the simplest case where you can write this as $x_3 = f(x_1, x_2)$, but the UC task tests the relation: $p(x_1, x_2 | x_3)$ and the FC task tests forecasting which are both complex relation that cannot be written down explicitly (and are also probabilistic).
>
> We chose to construct the synthetic variable using the form $x_3 = f(x_1, x_2)$ since it lets us test all these complex, implicit and probabilistic relations that appear in genuine real world data in an exact form.
>
> 2. Consider the OC task: where $x_3 = f(x_1, x_2)$, if $E_r$ is zero (no relational hallucination), then there is only one correct $x_3$ and hence you must predict accurately.
>
> For prompt-response style usage of foundation models (like in NLP), typical MSE measurements are actually not a great usecase. Consider the UC task: where $x_1, x_2 \sim p(x_1, x_2 | x_3)$, there is actually an infinite number of responses that are still relationally correct. Each dataset will itself have a distribution of values that are all correct. So which MSE do you use? This is indeed a strong supporting point for why the relational error is important and we will clarify this in the text.
>
> 3. We agree that we used a small simple model. We chose to use the simplest model possible that can demonstrate the point that we are trying to make: FMs hallucinate and here is a simple recipe for detecting and mitigating it.
>
> Indeed when scaling to longer or variable-length sequences you will have to change the architecture of the model you use. This is the case for any other algorithm - if you want to scale the dataset you apply a particular neural network to, you will most likely also want to use a bigger neural network. We have discussed this limitation in the 'Limitations and Further Work' in Section 5.
>
> 4. Our results do indeed suggest that current MVTS FMs are flawed, they can still be outperformed by small models and they hallucinate more than small models. This is an important result to raise.
>
> FMs are already being applied to MVTS, we are not the first to introduce large FMs to MVTS. We are presenting the current limits in their performance and suggesting new ways to evaluate and improve them.
>
> Writing Errors:
> 1. For $\hat{x}_{i}$ , the indices $i$ index the data dimension. The indices $i \in \mathcal{I}_r$ are the ones that have been imputed and are the complements of which indices you choose to provide as the prompt, which are denoted by the indices $i \in \mathcal{I}_p$.
>
> $\hat{x}_{i}$ is also used for the prompts because these are also imputed by the model (this is a property of rePaint algorithm), the model will return  $\hat{x}_i \approx x_i, \forall i \in \mathcal{I}_p$. The choice of this notation is intentional to highlight this subtlety.
>
> 2. Indeed the double hat notation is non-standard but there is a reason why we chose this notation as explained just above Equation 9. The double hat ($\hat{\hat{x}}$) denotes a ‘prediction’ of a target where the target is a previous prediction ($\hat{x}$). Of course we could choose another arbitrary symbol such as $\tilde{x}$ to represent this. We believe that this non-standard symbol makes sense, but will of course change this notation to an arbitrary one if the reviewer insists.
>
> 3. Multiple rows of results are bolded within the same column when the differences between them are not considered statistically significant and hence can be interpreted as a joint ‘best solution’. This is by design and is not a writing error.
>
> 4. We thank the reviewer for catching the mathematical typo for 'sampling $x_3$', we will correct that to $x_2$.

---

### Review · Reviewer_2WoA · 2026-04-28

**Summary Of Contributions:**

The paper studies hallucination in multivariate time-series imputation models, with a particular focus on what the authors call relational hallucination. The main technical idea is to use a diffusion-based score, CE, to estimate whether a model response is compatible with the prompt and the learned structure of the data. The paper then uses CE in two ways: first, to classify responses into different expected hallucination levels, and second, to filter multiple candidate responses by selecting the one with the lowest CE.

To evaluate this, the authors build relational versions of five standard MVTS datasets by adding one derived variable with a known relation to the others, so that relational error can be measured explicitly. They then test both a diffusion model trained on each dataset and two pretrained imputation foundation models, MOMENT and TIMER. I think the main strengths are that the paper tackles an interesting and underexplored problem, and that the relational hallucination framing is easy to understand. The main weaknesses are that the empirical setting is quite narrow, some of the headline numerical claims are framed too strongly relative to the actual tables, and the mitigation results for deterministic pretrained models rely on a sampling heuristic that is not well justified.

**Audience:**

Yes

**Audience Explanation:**

I think this paper would be of interest to people working on time-series foundation models, generative time-series modeling, and model reliability more broadly. The question it asks is a natural one: if MVTS models are used in a prompt-response setting, how can we tell when a response is inconsistent with the prompt or with learned relations among variables?
I also think the paper is interesting because it tries to translate a concern that is familiar in NLP, hallucination, into a time-series setting where the failure mode looks different. Even if readers disagree with parts of the framing, I think many of them would still find the relational viewpoint useful.

**Claims And Evidence:**

Yes

**Claims Explanation:**

I think the paper provides enough evidence for its main claims in the setting it actually studies. The experimental design is reasonable: the authors construct datasets where relational error can be computed directly, while CE itself is still produced by a learned diffusion model rather than by directly using the ground-truth relation. That makes it possible to test whether CE is actually tracking relational inconsistency.
The main empirical claims are also supported at a basic level by the results. Table 1 does show that the tested pretrained models can have substantial relational error, and Table 2 suggests that CE-based filtering often reduces relational error in this benchmark. I also appreciated that the appendix includes alternative diffusion-derived metrics instead of only showing the preferred one.
My reservations are mainly about scope and presentation rather than about whether the core empirical signal is there. The experiments are all in a fairly controlled three-variable setting with one injected or derived relation, so I do not think the paper should generalize too broadly beyond that. I also think the mitigation results for MOMENT and TIMER are less convincing than the detection results, since they depend on inference-time dropout without much justification or comparison to alternatives. So I am comfortable answering “yes,” but I do think the claims need to be phrased more carefully.

**Requested Changes:**

Critical changes：
1. The abstract and conclusion should present the quantitative results more carefully. The statement that pretrained MVTS FMs “relationally hallucinate on average up to 59.5% as much as a weak baseline” is technically supported by the table, but it foregrounds the average of the worse FM and risks sounding stronger than the underlying evidence. Table 1 also contains at least one case where TIMER is worse than the weak baseline (rWTH, OC, 1.03x baseline), which is not clearly discussed in the main text. Similarly, the mitigation claim of “up to 47.7%” emphasizes the single best case, while the average gain is much smaller. I would ask the authors to revise these headline claims so that best-case and average-case performance are not conflated.
2. The mitigation results for MOMENT and TIMER need stronger justification. Since these models are deterministic, the paper creates candidate samples by turning on dropout at inference time and then selecting the one with the lowest CE. This may be a useful heuristic, but the paper does not explain why this should be treated as a meaningful sampling scheme, nor does it compare against simpler alternatives. As a result, the mitigation results are promising but not yet as well supported as the paper suggests. At minimum this limitation should be discussed more directly, and ideally a control should be added.
3. The empirical scope should be stated more explicitly. The experiments are conducted on relational benchmarks that reduce to a three-variable setting with one injected or derived relation. This is a reasonable starting point, but narrower than the abstract currently suggests. Unless broader experiments are added, I think the scope of the claims should be tightened accordingly.
4. There is an inconsistency between the paper and code in the VPD relation. The paper describes humidity as a decimal fraction, while the implementation uses h / 100, i.e. humidity in percent, and also uses a slightly different constant. This does not appear to invalidate the experiments, but it should be corrected for reproducibility and consistency.
5. The appendix argues that CE is better than the alternative diffusion-derived metrics, but this comparison is only qualitative. Since this is the main ablation supporting the choice of CE, I think the paper should include at least one quantitative comparison, such as overlap coefficient or correlation with relational error.

Additional strengthening suggestions
6. A real-data qualitative example would help considerably. The toy heatmap in Figure 1 is intuitive, but it would be useful to also show a concrete example from one of the benchmark datasets where a response is relationally inconsistent with the prompt, together with its CE score.
7. The distinction from anomaly detection is well motivated conceptually, but an empirical comparison to a simple anomaly-detection baseline would strengthen the case further.
8. The limitations section would benefit from a clearer discussion of what parts of the method should transfer beyond the current MLP backbone and what parts may not.
9. There are also a few citation inaccuracies in the introduction that should be corrected.

---

> ### Author Response · Authors · 2026-05-04
>
> We thank the reviewer for taking the time to carefully and thoroughly read the work.
>
> 1. We agree with regards to the risk of sounding stronger that the underlying experimental results and will change this to the average value across all the results instead. We will also do this for the mitigation claim. We will also comment in the main text on the fact that FMs can even exceed the hallucination level of the weak baseline - which supports the point being made that FMs hallucinate extensively.
> 2. We found that turning on dropout and running the mitigation method surprisingly gave good results and serves as an extremely simple way to apply our mitigation method. We agree that there are no strong justification on why this should be the case, and will comment on this as a limitation.
> 3. We agree, we will add comments in the abstract to state that results were obtained using synthetically injected variables.
> 4. We thank the reviewer for spotting this, which required going through the codebase extensively. We will correct codebase to match paper’s text.
> 5. We will add tables containing the correlation coefficients.
>
> We will also find and correct the citation inaccuracies in the introduction and discuss the MLP backbone limitations.

---

### Review · Reviewer_ekzA · 2026-05-03

**Summary Of Contributions:**

This work defines the relational hallucination for MVTS hallucination, and propose a diffusion model to detect and estimate hallucination levels.

**Audience:**

Yes

**Audience Explanation:**

The work discusses the hallucination and time-series model, there might be audience have interest.

**Claims And Evidence:**

No

**Claims Explanation:**

1.When the authors define 'relational hallucination', as a definition, please clearly provide the following:

1) What is the function exactly?
2) What are the domains and properties of the function?
3) What are the conditions required?
4) How is it consistent with the existing definition?

Without the ingredients, it is hard to evaluate the well-definedness of the definition.

2. There is no or very weak justification of using Diffusion model for detecting the hallucination.

3. Why the L2 distance can be used for evaluating the relational hallucination of the prediction? Back to the first question, how is the relational hallucination defined? If the definition is not even clear, how could one evaluate the optimality of the model that is used to detect relational hallucination? With that being said, even the minimum RMSE is achieved, is the relational hallucination resolved?

**Requested Changes:**

Please see the comments above.

---

> ### Author Response · Authors · 2026-05-04
>
> This is a misunderstanding of the text.
>
> 1: As discussed in the second paragraph of Section 2, relations between variables can be represented using the following form $f(x_1, …, x_n) = 0$. Note that this relation can even be probabilistic. Each MVTS dataset will have it’s own ground truth relation (which are typically not known). For instance a dataset consisting of $x_1$ = temperature, $x_2$ = humidity and $x_3$ = VPD (a nonlinear function of temperature and humidity) will be related according to the function  $x_3 - VPD(x_1, x_2) = 0$, as highlighted in the ‘Relational Variables’ paragraph in Section 4. All the relations for all the datasets used are also defined in this paragraph.
>
> As discussed in the first two paragraphs of Section 2, relational hallucination is a subset of the concept of ‘out of distribution’. With regards to NLP, this is analogous to when the response is irrelevant to the prompt or when there are self-contradictions.
>
> 2: Diffusion models serve as a good candidate for modelling the relations since they have been shown to be highly expressive. For $f(x_1, …, x_n) = 0$ the valid solutions form a set/manifold $M = ${$ x : f(x)=0 $}. A diffusion model can learn to generate samples concentrated on this solution set: $p(x_1, …, x_n ∣ f(x)\approx0)$. This makes them a good candidate for learning such relations. As discussed in the first paragraph of Section 3, prior works have shown that diffusion models can indeed detect hallucination. This makes them a good candidate for detecting hallucination.
>
> 3: As discussed in the first paragraph of Section 2, the relations within a dataset can be written down as $f(x) = 0$, a measure of how much this relation is broken can therefore be measured using $E_r=|f(x)|$. Indeed if $E_r=0$ or $E_r^2=0$, then there is no relational hallucination.

---

> > ### Comment · Reviewer_ekzA · 2026-05-18
> > **Response**
> >
> > Thank you for your response. Below are my followup questions.
> >
> > 1. The authors are expected to answer the questions on what is the $definition$ of relational hallucination. Simply take $x_3 - VPD(x_1, x_2) = 0$ as an example is NOT a definition. Can you elaborate more on 'can even be probabilistic'?
> >
> > 2. Diffusion model can be a good candidate for generation. What I was trying to understand is its justification to be used in the studied setting. A minor comment: If the variables are not considered to be probabilistic before, where is the randomness from in $p(x_1, …, x_n ∣ f(x)\approx0)$?
> >
> > 3. What is the 'measure' exactly? $f(x) = 0$ is an arbitrary function without clear explicit expression.

---

> > > ### Author Response · Authors · 2026-05-19
> > >
> > > 1)
> > > The definition of relational hallucination is defined in the second paragraph of Section 2. Explicitly, please see the following quote from the text:
> > >
> > > “A relation between a set of $N$ variables $x = ${$x_1, x_2, ..., x_N $} can
> > > be written as $f(x) = 0$, where $f$ is some ground-truth function that defines the relation. The ‘relational error’ which measures the degree of which the relation is broken can then be defined as $E_r := |f(x)|$. Relational hallucination can then be defined as the case when the model returns a set of variables that has ‘high’ relational error, relative to some threshold.”
> > >
> > > To put it concretely, consider the relation defined by $f(x_1, x_2, x_3) = x_3 - VPD(x_1, x_2) = 0$. For this particular example, relational hallucination occurs when $E_r = |x_3 - VPD(x_1, x_2)|$ is greater than some threshold. This threshold should be set as is relevant to each context. For simplicity, we use a quantile-based threshold given the training set.
> > >
> > > What we mean by ‘this can even be probabilistic’ means that the methods proposed can handle probabilistic cases, for instance a relation defined by something of the form $p(x_3∣x_1,x_2)$, since diffusion models naturally handle probability distributions.
> > >
> > > 2.
> > > As discussed in the first paragraph of Section 3, prior works have shown that diffusion models can indeed detect hallucination. This makes them a good candidate for detecting hallucination. Since diffusion models were applied in prior works for hallucination detection in other domains, diffusion models are good candidates to be applied in this work which is to detect hallucination in MVTS.
> > >
> > > From a more intuition-based view, diffusion models learn probablity distributions around the training data, and are highly expressive. They learn to iteratively push incorrect samples (eg. pure noise) towards areas with high probability (given the training data). These areas are where the relations are 'correct', given the training data. Hence a measure based on how far diffusion models 'push' a candidate response is a good candidate for detecting hallucination.
> > >
> > > If the relations in the dataset is not probabilistic (which in real settings is highly unlikely due to noise), diffusion models can still be applied and the learnt distribution will just be very sharp around the data.
> > >
> > > 3.
> > > We believe this is a misunderstanding and is clarified by the above two points.

---

### Review · Reviewer_7su8 · 2026-05-09

**Summary Of Contributions:**

This submission studies hallucination in multi-variate time-series imputation foundation models. The paper proposes two definitions: distributional hallucination, where the prompt-response pair is out of distribution with respect to a target dataset, and relational hallucination. The main technical proposal is to train a diffusion model on the target time-series domain and use a Combined Error (CE) reconstruction-style metric as a proxy for relational hallucination. The authors then use quartiles of CE for hallucination-level detection and sample-and-filter responses by choosing the candidate with lowest CE for mitigation. Experiments add relational variables to five common MVTS datasets and evaluate a weak mean baseline, an in-domain diffusion model, MOMENT, and TIMER on over-constrained, under-constrained, and forecasting-style prompts.

**Audience:**

Yes

**Audience Explanation:**

The paper addresses an important and underexplored problem: reliability evaluation for time-series foundation models. The framing of hallucination for MVTS imputation is interesting, and a black-box consistency detector could be useful to researchers in time-series modeling, diffusion models, uncertainty estimation, and trustworthy ML.

**Claims And Evidence:**

Yes

**Claims Explanation:**

The paper provides evidence that CE correlates with relational errors on the authors’ constructed benchmarks, and CE-based filtering can reduce relational error. However, the evidence is not yet fully convincing for the broader claim of general hallucination detection and mitigation in MVTS FMs.

Weaknesses:

1. The relational benchmarks are mostly synthetically constructed from deterministic algebraic relations, so it is unclear whether the method transfers to natural, noisy, or high-dimensional real-world constraints.
2. The baselines are too limited. CE should be compared against simpler reconstruction-error methods, autoencoders, anomaly/OOD detectors, likelihood/score-based diffusion metrics, and uncertainty-based filtering.
3. Detection is evaluated mainly with low/high overlap coefficients. More standard metrics such as AUROC, AUPRC, calibration, and correlation with true relational error are needed.
4. Mitigation only reports relational-error reduction. The paper should also report imputation/forecasting accuracy, since the lowest-CE sample may be relation-consistent but inaccurate.

**Requested Changes:**

Please refer to weaknesses.

---

> ### Author Response · Authors · 2026-05-20
>
> 1. Since the diffusion model does not assume the form of the relations, the model has no access to the chosen relation structure, which is the form $x_3 - g(x_1, x_2) = 0$, where $g(.)$ is a simple algebraic function. The diffusion model therefore needs to learn the prior $p(x_1, x_2, x_3)$. We explicitly test for cases where the the relations are complicated and cannot be written down in closed form, and are also probabilistic using the different tasks. Indeed the OC task, is simple since the relations if $x_3 = g(x_1, x_2)$. The UC task however is of the form $p(x_1, x_2 | x_3)$, which is probablistic (for a given $x_3$ there are multiple correct $x_1$ and $x_2$) and cannot be written down as a simple algebraic function. The same is also true for the FC task.
> 2. We will compare against other diffusion metrics for hallucination detection from literature.
> 3. We will add correlation coefficients with groundtruth relational error in a figure.
> 4. We will add an additional table that reports the imputation error showing that it remains statistically the same after the mitigation procedure.

---

### Decision · Action_Editor_HEBo · 2026-06-24

**Recommendation:** Accept with minor revision

**Audience:**

Yes

**Audience Explanation:**

At least some individuals in TMLR's audience would be interested in the findings of this paper. The paper addresses an emerging and underexplored reliability issue for multivariate time-series foundation models: how to characterize, detect, and mitigate hallucination-like failures in prompt-response-style time-series imputation. This topic is relevant to researchers working on time-series foundation models, generative time-series modeling, diffusion models, out-of-distribution detection, anomaly detection, uncertainty estimation, and trustworthy machine learning.

The proposed framing of relational hallucination is also potentially useful because it translates the broader concern of hallucination into a setting where inconsistencies can be operationalized and empirically measured. Even though the current evidence is mainly based on controlled relational benchmarks with injected or derived relations, the problem formulation and empirical observations are likely to stimulate further discussion and follow-up work in the TMLR community.

Therefore, I believe the submission satisfies the TMLR audience-interest criterion.

**Claims And Evidence:**

Yes

**Claims Explanation:**

The submission provides sufficient evidence to support its main claims within the specific setting studied. The paper introduces a clear operationalization of relational hallucination for multivariate time-series foundation models and evaluates the proposed diffusion-based CE score on constructed relational benchmarks where the ground-truth relational error can be directly measured. The experiments show that CE correlates with relational inconsistency and that CE-based filtering can reduce relational error in the evaluated models and datasets.

I also note that the revised manuscript appears to have addressed several important concerns raised during review, including clarifying the empirical scope, rephrasing headline quantitative claims more carefully, discussing the limitations of inference-time dropout for deterministic models, correcting reproducibility issues, and adding further quantitative comparisons for the CE metric.

That said, the evidence mainly supports the claims in the controlled three-variable relational benchmark setting with injected or derived relations. The paper should avoid overgeneralizing the results to all forms of hallucination in real-world MVTS foundation models. Overall, after the revisions, I consider the core claims to be supported by accurate, convincing, and reasonably clear evidence, provided the scope is interpreted appropriately.

**Resubmission Of Major Revision:**

The authors may consider submitting a major revision at a later time.